# The MEK1/2 Inhibitor ATR-002 (Zapnometinib) Synergistically Potentiates the Antiviral Effect of Direct-Acting Anti-SARS-CoV-2 Drugs

**DOI:** 10.3390/pharmaceutics14091776

**Published:** 2022-08-25

**Authors:** André Schreiber, Benjamin Ambrosy, Oliver Planz, Sebastian Schloer, Ursula Rescher, Stephan Ludwig

**Affiliations:** 1Institute of Virology (IVM), Centre for Molecular Biology of Inflammation, University of Muenster, 48149 Muenster, Germany; 2Interfaculty Institute for Cell Biology, Department of Immunology, Eberhard Karls University Tuebingen, Germany and Atriva Therapeutics GmbH, 72072 Tuebingen, Germany; 3Research Group Regulatory Mechanisms of Inflammation, Institute of Medical Biochemistry, Centre for Molecular Biology of Inflammation, University of Muenster, 48149 Muenster, Germany; 4Leibniz Institute for Experimental Virology, 20251 Hamburg, Germany; 5Interdisciplinary Centre of Clinical Research (IZKF), Medical Faculty, University of Muenster, 48149 Muenster, Germany

**Keywords:** SARS-CoV-2, COVID-19, antiviral drug, MEK1/2 inhibitor, ATR-002, drug synergy, Molnupiravir, Remdesivir, Nirmatrelvir, Ritonavir, Paxlovid

## Abstract

The coronavirus disease 2019 (COVID-19) represents a global public health burden. In addition to vaccination, safe and efficient antiviral treatment strategies to restrict the viral spread within the patient are urgently needed. An alternative approach to a single-drug therapy is the combinatory use of virus- and host-targeted antivirals, leading to a synergistic boost of the drugs’ impact. In this study, we investigated the property of the MEK1/2 inhibitor ATR-002’s (zapnometinib) ability to potentiate the effect of direct-acting antivirals (DAA) against SARS-CoV-2 on viral replication. Treatment combinations of ATR-002 with nucleoside inhibitors Molnupiravir and Remdesivir or 3C-like protease inhibitors Nirmatrelvir and Ritonavir, the ingredients of the drug Paxlovid, were examined in Calu-3 cells to evaluate the advantage of their combinatory use against a SARS-CoV-2 infection. Synergistic effects could be observed for all tested combinations of ATR-002 with DAAs, as calculated by four different reference models in a concentration range that was very well-tolerated by the cells. Our results show that ATR-002 has the potential to act synergistically in combination with direct-acting antivirals, allowing for a reduction in the effective concentrations of the individual drugs and reducing side effects.

## 1. Introduction

Since its emergence in 2019, severe acute respiratory syndrome coronavirus-2 (SARS-CoV-2) has rapidly spread all over the world, causing millions of cases of the coronavirus disease 2019 (COVID-19) [1]. While in many cases infections are asymptomatic or manifest through mild upper respiratory symptoms, there is a significant number of cases exhibiting symptomatic pneumonia or severe acute respiratory distress syndrome (ARDS), requiring hospitalization and even intensive medical care [2]. The ongoing pandemic not only poses a major health burden to humans worldwide but in addition strains national health systems on a global scale. The worldwide vaccination campaign improves the protection against COVID-19 for the vaccinated individuals and is a very effective prophylactic anti-SARS-CoV-2 measure [3]. Nevertheless, vaccination efficiency is limited by the availability of vaccine doses per country, the willingness of the population to get vaccinated and possible emerging escape mutations [3,4,5]. In addition to immunization as a prophylactic countermeasure, antiviral and immunomodulatory agents are of great therapeutic importance, especially for the treatment of patients suffering from severe COVID-19 [6]. Antiviral drugs can impede the viral spread within the patient, while immunomodulatory agents alleviate the symptoms of severe COVID-19 [2]. Because the development of novel antiviral drugs is time-consuming, drugs that are already clinically licensed are tested for their efficacy against COVID-19. Those substances can either target viral components or cellular mechanisms that are exploited by the pathogen during a viral infection. Direct-acting antivirals (DAA) can be very efficient; however, such a treatment bares the risk of provoking resistance by introducing mutations, e.g., the rapid emergence of influenza viruses resistant against the M2 ion channel inhibitor, Amantadine [7]. In comparison, host-targeted antivirals (HTA) are less prone to forcing mutations but might exhibit a lower inhibitory effect on the virus. To minimize the pressure of forced escape mutations and maximize the inhibitory potency, combinational therapies with synergistic antiviral effects are a promising approach [8,9,10].

Previous studies have shown that several RNA viruses, including SARS-CoV-2, misuse the cellular Raf/MEK/ERK signaling pathway to promote replication, and accordingly, the inhibition of the pathway with MEK1/2-specific inhibitors impedes viral replication [7,11,12,13,14,15,16,17]. ATR-002 (zapnometinib, PD 0184264), a small molecule inhibitor of the two isoforms MEK1 and MEK2, was previously reported to have antiviral properties against influenza viruses and SARS-CoV-2 [11,18], demonstrating that ATR-002 is an HTA candidate. Interestingly, ATR-002 appears to act against acute respiratory virus infections using a dual effect. On one hand, the compound interferes with the replication process and thereby leads to the reduction in influenza virus or SARS-CoV-2 titers; on the other hand, the drug results in the downmodulation of virus-induced expression of cytokines and chemokines that are causative for the hyperinflammatory cytokine storm at late stages of viral diseases such as COVID-19 [11,18].

Here, we demonstrate that combined treatments of ATR-002 and different SARS-CoV-2 directed antivirals, such as the inhibitors of viral RNA-dependent RNA-polymerases (RdRp) Molnupiravir [19] and Remdesivir [20] or the 3C-like protease inhibitors Nirmatrelvir [21] and Ritonavir [22,23], combined known as Paxlovid, display potentiated antiviral efficacy because of drug synergy.

## 2. Materials and Methods

### 2.1. Cell Lines

The human airway epithelial cells (Calu-3) were cultured in Dulbecco’s modified Eagle medium/Nutrient Mixture F12-Ham (DMEM/F12-HAM) (Sigma-Life Science, St. Louis, MO, USA) with 10% (*v*/*v*) fetal bovine serum (FBS) (Capricorn Scientific, Ebsdorfergrund, Germany) and 1% (*v*/*v*) Penicillin/Streptomycin (P/S) (Sigma-Life Science). African green monkey kidney epithelial cells (VeroE6) were cultured in DMEM supplemented with 10% (*v*/*v*) FBS. Vero76 cells stably overexpressing TMPRSS2 (Vero76-TMPRSS2, German Primate Center Goettingen, Goettingen, Germany) were cultured in DMEM supplemented with 10% (*v*/*v*) FBS, 1% (*v*/*v*) P/S and 10 µg/mL blasticidin (Roth). Cells were incubated under humidified conditions at 37 °C and 5% CO_2_.

### 2.2. Viruses

All SARS-CoV-2 experiments were performed in a laboratory approved for biosafety level (BSL) 3 work. Viruses were isolated as described previously [11]. In order to exclude genomic mutations due to cell culture propagation, virus stocks were sequenced before usage. Sequences can be accessed at GISAID.org: D614G: hCoV-19/Germany/FI1103201/2020; München-01: München-1/02/2020/984; α-B1.1.7: hCoV-19/Germany/NW-RKI-I-0026/2020; β-B1.351: hCoV-19/Germany/NW-RKI-I-0029/2020; δ-B1.617.2: hCoV-19/Germany/326763/2021.

### 2.3. Virus Propagation

SARS-CoV-2 viruses were propagated on Vero76-TMPRSS2 cells in DMEM with 2% (*v*/*v*) FBS, 1% (*v*/*v*) P/S, 1% (*v*/*v*) sodium pyruvate solution (Gibco, Waltham, MA, USA), 1% (*v*/*v*) non-essential amino acid (NEAA) solution (Sigma-Life Science) and 1% (*v*/*v*) HEPES (1 M; pH 7.2) (Sigma-Aldrich, St. Louis, MO, USA) using an MOI of 0.01. After 72 h post-infection, the virus-containing supernatants were collected, and the virus titers were determined by plaque titration.

### 2.4. Virus Infection

Virus dilutions were prepared in cell culture medium. A total of 1 × 10^6^ Calu-3 cells seeded in 12 wells were washed once with PBS and infected for 1 h at 37 °C, followed by a second PBS wash step and incubated in cell culture medium supplemented with the different inhibitors. DMSO served as a control and was used in a final concentration of 0.1%.

### 2.5. Inhibitors

All inhibitors were dissolved in DMSO (Roth). ATR-002 (zapnometinib, PD 0184264) was obtained from Atriva Therapeutics GmbH (Tuebingen, Germany). Remdesivir (Veklury^®^) and Ritonavir (Norvir^®^) were purchased from Tocris (Wiesbaden-Nordenstadt, Germany). Molnupiravir (Lagevrio^®^) and Nirmatrelvir (PF-07321332) were purchased from Selleckchem (Planegg, Germany).

### 2.6. Virus Titration by Plaque Assay

SARS-CoV-2 solutions were diluted 10-fold in PBS supplemented with 0.01% (*w*/*v*) CaCl_2_ (Roth), 0.01% (*w*/*v*) MgCl_2_ (Roth), 0.6% (*v*/*v*) bovine serum albumin (BSA) (35%) (Sigma-Aldrich) and 1% (*v*/*v*) P/S. Confluent monolayers of VeroE6 cells were infected for 1 h at 37 °C with the dilution series and covered with plaque medium (2% (*v*/*v*) FBS, 35% (*v*/*v*) agar (2%) (Oxoid) and 63% (*v*/*v*) 2× MEM (1% (*v*/*v*) 100× Penicillin/Streptomycin/L-Glutamine solution (10,000 U/mL Penicillin; 10,000 µg/mL Streptomycin; 29.2 mg/mL L-Glutamine) (Gibco), 1.2% (*v*/*v*) BSA (35%), 2% (*v*/*v*) HEPES, 3.2% (*v*/*v*) NaHCO_3_ (7.5%) (Gibco), 20% (*v*/*v*) 10× MEM (Gibco))). Plaque forming units were analyzed after 72 h incubation at 37 °C.

### 2.7. Cell Cytotoxicity Assay

The colorimetric MTT assay was used to evaluate the metabolic activity as indicator of the activation, proliferation and cytotoxicity [24]. Calu-3 cells were incubated with the inhibitors in a mono- or double-treatment using the indicated concentrations for 48 h. Staurosporine solution (1 µM) (Sigma) served as a positive control for cytotoxic effects. Following 48 h of treatment, MTT (3-(4,5-dimethylthiazol-2-yl)-2,5-diphenyltetrazolium bromide) (1 mg/mL) (Sigma) was added to the cells for 3 h. The supernatant was aspirated, DMSO was added to the cells for 10 min and the OD_562_ was subsequently measured using an Epoch microplate spectrophotometer (BioTek, Winooski, VT, USA) to evaluate the cell viability according to the manufacturer’s protocol (Sigma).

### 2.8. Quantification and Statistical Analysis

The sample size required to detect 90% reduction in viral titers at powers > 0.8 and a significance level of 0.05 was determined using G*Power 3.1 (RRID: SCR_013726) and a priori power analysis [25]. Data were analyzed and graphically visualized using GraphPad PRISM Version 8.0 (RRID: SCR_002798). The open-source web application SynergyFinder Plus [26] was used to evaluate and visualize the drug combinatory screening on the basis of the Bliss independence model (Bliss), highest single-agent model (HSA), Loewe additivity model (Loewe) and the Bliss and Loewe combined Zero Interaction Potency model (ZIP) [27,28,29]. The reference of the Bliss independence model expects that the two drugs evoke independent effects. The highest single-agent model assumes that the reference drug combinational effect equals the maximal effect of a single drug. The Loewe additivity model defines the reference as an effect of a drug combined with itself. The Zero Interaction Potency model compares the changes in the potency of the dose–response curves of single drugs with their combinations, assuming no changes in the response curves if the drugs do not interact [28].

## 3. Results

### 3.1. Single Treatment of ATR-002 or Direct-Acting Antivirals (DAA) Efficiently Inhibits Replication of SARS-CoV-2

We recently evaluated the potential of MEK1/2 inhibitor zapnometinib, here designated as ATR-002, as a drug candidate against COVID-19 based on its direct inhibitory effect on the SARS-CoV-2 replication cycle along with a secondary beneficial inhibitory effect on the release of proinflammatory cytokines and chemokines [11]. While monotherapy with ATR-002 already significantly reduced viral titers, we wanted to assess whether antiviral activity could be potentiated by combinatory treatment of ATR-002 with DAAs including Molnupiravir (MPV), Remdesivir (RDV), Nirmatrelvir (NTV) and Ritonavir (RTV). To define drug concentrations that are suitable for the synergistic evaluation, effective inhibitory concentrations (EC_10_, EC_50_, EC_90_) of all drugs were determined, representing a 10%, 50% or 90% inhibitory effect on viral replication. Additionally, we calculated the EC_1_ concentrations. As these concentrations do not have an impact on viral propagation, they were used as a defined concentration, which does not alter the viral replication when used in a monotherapy. Calu-3 cells were infected with the D614G SARS-CoV-2 isolate hCoV-19/Germany/FI1103201/2020 using a multiplicity of infection (MOI) of 0.01, followed by a treatment of the respective drugs 1 h post-infection (h.p.i.) for a total incubation time of 48 h. In line with our recently published work [11], we found a dose-dependent reduction in SARS-CoV-2 titers after ATR-002 treatment, allowing for the calculation of the above-mentioned EC values. Additionally, we could determine EC values for MPV, RDV, NTV and RTV (Figure 1 and Appendix A), which are comparable to recently published data [10,30,31,32,33]. Of note, the calculated concentrations required for the antiviral effect are non-cytotoxic for all validated drugs in Calu-3 cells after 48 h treatment (Appendix A), indicating that the inhibitory effects are not aberrantly caused by impaired cell viability.

### 3.2. Synergistic Drug Interactions between ATR-002 and DAAs against SARS-CoV-2

We next investigated the synergistic effects of the DAAs MPV, RDV, NTV and RTV in combination with the HTA ATR-002. Calu-3 cells infected with SARS-CoV-2 (MOI 0.01) were either treated with the respective inhibitors as a single treatment (drug + DMSO) or a combinatory treatment (drug_1_ + drug_2_). Several drug combinations of all tested DAAs (MPV, RDV, NTV, RTV) with ATR-002 showed an overall combinatory effect that by far exceeds the sum of the respective monotherapies (Figure 2, Appendix A). Of note, the strong antiviral effects are caused by the inhibitory actions of the drugs and are not the result of enhanced cell cytotoxicity (Appendix A). To analyze the drug synergy of the drug pairs, we used the open-source web application SynergyFinder Plus. Four different reference models (Bliss, HSA, Loewe, ZIP) were used to quantify the degree of interactions (synergy score) and the overall treatment efficacy (combination sensitivity score; CSS). Examination of the drug interactions and landscape visualizations revealed strong synergistic effects for ATR-002 combined with all tested DAAs (Figure 3 and Appendix A). For all reference models, the highest synergy values were found for the combinations MPV 0.047 µM, RDV 0.034 µM, NTV 0.006 µM or RTV 0.057 µM when administered with 12.05 µM ATR-002, indicating the robustness of the data (Appendix A). All tested reference models showed positive overall synergy scores for the different drug combinations, confirming the average synergistic mode of actions (Appendix A). To compare the synergistic potency of ATR-002 with a treatment strategy that only targets the virus (DAA_1_ + DAA_2_), we further tested the combination of the viral RdRp-inhibitor RDV with the 3C-like protease inhibitor NTV. Interaction determination revealed synergisms if NTV was used in higher concentrations (NTV: 0.096 µM, 2.152 µM) combined with all tested RDV concentrations. However, surprisingly lower NTV concentrations (NTV: 0.0002 µM, 0.006 µM) led to an antagonistic effect (Figure 2e, Figure 3e and Appendix A). In line with these results, the overall synergy scores, as well as the CSS, were lower compared to the HTA + DAA treatments (Appendix A).

### 3.3. Strong Synergistic Effects of ATR-002 Combinations with DAAs against Different SARS-CoV-2 Variants

Because of the strong synergy of the HTA + DAA drug pairs, we further evaluated the effect of the drug combinations on viral replication for other SARS-CoV-2 variants. For this purpose, the drug combinations with the highest synergy values detected for the D614G isolate (MPV: 0.047 µM/ATR-002: 12.05 µM; RDV: 0.034 µM/ATR-002: 12.05 µM; NTV: 0.006 µM/ATR-002: 12.05 µM; RTV: 0.057 µM/ATR-002: 12.05 µM) were chosen and tested against the Wuhan strain of the virus (München-01), α-B1.1.7, β-B1.351 and δ-B1.617.2 (Figure 4 and Appendix A). Overall reduction in viral titers after single-drug treatments was comparable between the different VOCs α-B1.1.7, β-B1.351 and δ-B1.617.2 and stronger for the Wuhan strain, which is in line with recently published data [34]. All drug combinations reduced viral titers of the different variants far beyond an additive effect of the single treatments (Figure 4 and Appendix A). The most pronounced differences between the investigated combinational inhibitory effect and a calculated additive effect of the titer inhibition were found for ATR-002 + MPV (β-B1.351, δ-B1.617.2), ATR-002 + RDV (α-B1.1.7) or ATR-002 + NTV (München-01), indicating that these combinations might exhibit the highest levels of synergy.

## 4. Discussion

The global health challenge caused by the SARS-CoV-2 pandemic has led to the identification of several drug candidates and treatment strategies, either interfering with virus replication itself or attenuating COVID-19 disease progression [11,32]. Drug combination is a helpful strategy to increase the effect of treatment and lower the risk of an obliged resistance to any of the drugs. Furthermore, drug combinations often allow for lower single-drug doses, thereby preventing the possibility of unwanted side effects [35,36]. In the past, synergistic effects could not only be shown for inflammation and cancer therapy [36,37] but also for the treatment of viral infections, including influenza virus (IV) [38,39], hepatitis C virus (HCV) [40], human immunodeficiency virus (HIV) [41], poliovirus (PV) [42] or Ebola virus (EV) [43]. In light of the ongoing pandemic, there is a constant demand for efficient drugs against COVID-19; however, the therapeutic potential of drug combinations that act synergistically in decreasing viral titers and may alleviate the symptoms of an acute SARS-CoV-2 infection has only poorly been evaluated so far. The first studies on drug synergies of host-targeted antivirals (HTAs) with direct-acting antivirals (DAAs) yielded quite promising results [9,10,30].

In a previous study, we already confirmed that successful SARS-CoV-2 infection relies on the activation of the cellular Raf/MEK/ERK signaling pathway. The inhibition of the kinase cascade with the MEK1/2-specific inhibitor ATR-002 (zapnometinib) resulted in a decreased cellular internalization, and thus a reduction in the production of progeny virus particles in a virus-variant-independent manner [11]. The antiviral effect of ATR-002 together with its auspicious bioavailability [18] and safety profile, as evidenced in a phase I clinical trial (ClinicalTrials.gov Identifier: NCT04385420), suggested ATR-002 as a promising candidate for the treatment of SARS-CoV-2 infections. Currently, the anti-SARS-CoV-2/COVID-19 properties of ATR-002 are under evaluation in a phase II clinical trial (RESPIRE: (ClinicalTrials.gov Identifier: NCT04776044)). Nevertheless, HTAs might only impair viral infections rather than eliminate the virus, while DAAs are prone to provoke resistances [7,8,9,10]. Combinations of HTAs and DAAs have the potency to increase the treatment window against upcoming SARS-CoV-2 variants, which might display decreased susceptibility against HTAs and DAAs, to eradicate the pathogen more efficiently. Additionally, drug combinations exhibiting synergistic properties allow for the use of reduced amounts of the respective drugs, even down to concentrations that would not be effective in single-drug doses. This might also reduce the likeliness of potential adverse effects. The present study is focused on the ability of ATR-002, a prototype HTA, to act synergistically with DAAs such as the nucleoside analog RdRp-inhibitors Molnupiravir and Remdesivir, as well as the 3C-like protease inhibitors Nirmatrelvir and Ritonavir. While recent studies have already shown that Remdesivir can act synergistically but also antagonistically with other DAAs in combinational therapy against coronaviruses [43,44,45], comparable results for combinations of HTAs and Remdesivir are scarce [9,10,30] or even completely missing for other DAAs, such as Molnupiravir, Nirmatrelvir and Ritonavir. To our knowledge, this is the first study to report synergistic properties of Molnupiravir, Nirmatrelvir and Ritonavir in general, as well as for MEK1/2 inhibitors used as SARS-CoV-2 antivirals. The combinatory effect of ATR-002 and DAAs shown in this study goes far beyond the additive effects of the monotherapies, indicating a synergistic mode of action for the respective drug pairs. While the exact mechanism of the here-described synergistic property of ATR-002 with several different anti-SARS-CoV-2 drugs is still unknown, one could hypothesize that the drug combination may sensitize SARS-CoV-2 to either of both drugs. For example, a partial inhibition of internalization by ATR-002 may reduce the virus load within the cell to a level that may be more efficiently targeted by the downstream action of polymerase or protease inhibitors, even in lower concentrations. Along that line, a 50% reduction in ERK1/2 activation achieved with 10 µM concentration of ATR-002, which alone did not show an antiviral effect [11], may be sufficient to inhibit viral replication when combined with another drug. Furthermore, despite the primary effect on internalization, moderate reductions in viral titers were also found when ATR-002 treatment was initiated 2 h.p.i., indicating an additional replication-supportive role of the Raf/MEK/ERK pathway within the SARS-CoV-2 life cycle [11]. This supportive function can putatively be compensated when MEK1/2 is inhibited alone but might contribute to the synergistic effect evaluated in this study. Moreover, co-administration of substances changing a drug’s metabolism can increase the therapeutic benefit, as described for Paxlovid. In addition to its function as a protease inhibitor, Ritonavir additionally inhibits the CYP3A-mediated metabolic degradation of Nirmatrelvir, resulting in a decreased turnover of Nirmatrelvir [46]. Comparably, the here-analyzed substance combinations might interfere with the drug’s metabolism, leading to prolonged periods of action. One should, however, state that all of these proposed mechanisms are still hypothetical and are still lacking experimental proof.

During the ongoing pandemic, several new virus variants emerged. Hence, drugs possessing antiviral properties against a broad spectrum of SARS-CoV-2 variants are favorable, as they might also have potent inhibitory effects against new and upcoming variants. We could already show that ATR-002 effectively inhibits SARS-CoV-2 variants of concern, including α-B1.1.7, β-B1.351 and δ-B1.617.2 [11]. In line with these results, the inhibitory effect of the drug combinations used in this study on VOCs surpassed the sum of the monotherapeutic effects, comparable to the effects shown for the D614G variant, indicating a variant-independent synergistic impact on SARS-CoV-2 viruses. Therefore, ATR-002 might be a beneficial supplement to boost the effect of DAA therapies. An advantage of an HTA is the reduced possibility of the emergence of resistance-introducing mutations because the virus cannot replace the missing cellular mechanism. For further studies, it would be interesting to test if a combinational treatment would also lower the risk of emergence of resistances.

Apart from the synergistic benefit of the drug combination, it is important to consider possible side effects caused by the combinatory application in patients. So far, no adverse drug–drug interactions caused by a combinational treatment are reported for Molnupiravir, other than for Remdesivir and Nirmatrelvir/Ritonavir (Paxlovid) [47]. While the combination of ATR-002 with these DAAs improved the antiviral effect compared to a single treatment in vitro, it is necessary to conduct in vivo experiments, not only because the antiviral effects might differ between in vitro and in vivo systems but also to exclude possible negative drug–drug interaction effects caused by the combinational treatment. Furthermore, the exact drug doses used in patients should be carefully adjusted according to their pre-existing conditions. However, the synergism of the DAAs in combination with ATR-002 is detectable even at the lowest tested concentrations, enabling a pharmacological treatment window in which major side effects might not occur.

## 5. Conclusions

The MEK1/2 inhibitor ATR-002 (zapnometinib) does not only act as a stand-alone drug to inhibit SARS-CoV-2 replication and virus-induced cytokine expression but also acts strongly synergistic with established DAAs, such as Molnupiravir, Remdesivir, Nirmatrelvir and Ritonavir. Such drug combinations may help to decrease effective concentrations of each individual drug, thereby reducing the likeliness of adverse events. Disease-stage-tailored combination of an HTA, comprising both antiviral and immunomodulatory features, with a DAA may also enhance the effectiveness of treatment throughout all stages of the COVID-19 disease.

## Figures and Tables

**Figure 1 pharmaceutics-14-01776-f001:**
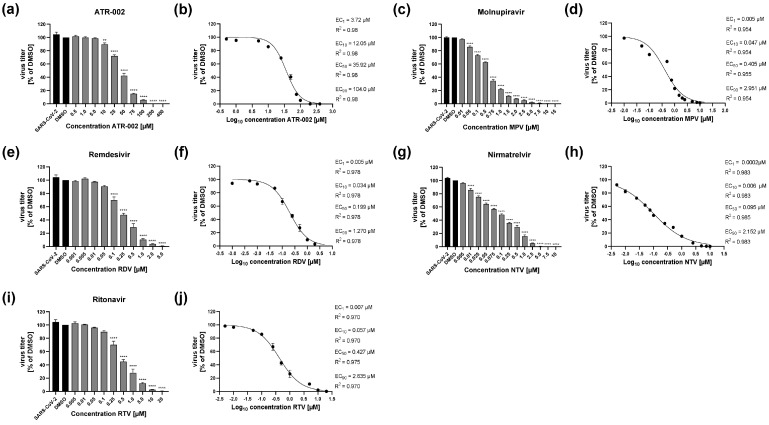
Inhibitory effect of ATR-002, MPV, RDV, NTV and RTV on SARS-CoV-2 replication. Calu-3 cells were infected with SARS-CoV-2 (D614G) (MOI: 0.01). One h.p.i. drug treatment was initiated. Untreated (SARS-CoV-2) and DMSO-treated cells served as negative controls. Viral titers were analyzed 48 h.p.i. (**a**,**c**,**e**,**g**,**i**). Viral titers in % of DMSO. DMSO was arbitrarily set to 100%. Shown are means ± SEM of five independent experiments, each performed in triplicates. Significance was calculated by one-way ANOVA in combination with a Dunnett’s multiple comparisons test using DMSO as reference (** *p* ≤ 0.0021; **** *p* ≤ 0.0001) (see also Appendix A). (**b**,**d**,**f**,**h**,**j**) Calculation of the EC values.

**Figure 2 pharmaceutics-14-01776-f002:**
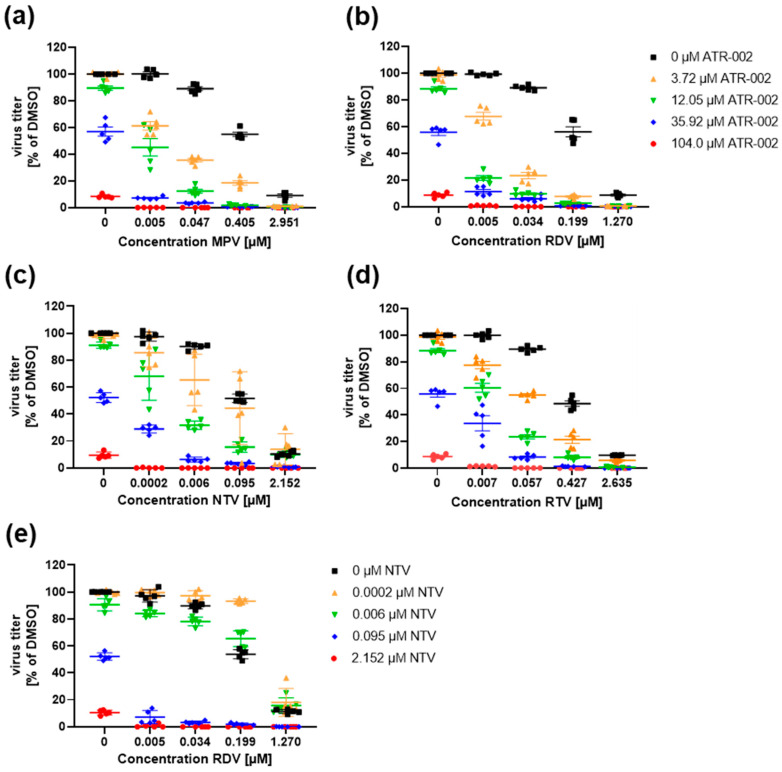
Antiviral effect of the combinational treatments in Calu-3 cells. Calu-3 cells were infected with SARS-CoV-2 (D614G) (MOI: 0.01). Combinational drug treatments were initiated 1 h.p.i. Viral titers were analyzed 48 h.p.i. Shown are means ± SEM of five independent experiments, each performed in triplicates. (**a**–**d**) DAAs (MPV, RDV, NTV, RTV) in combination with the HTA ATR-002. (**e**) Combination of the DAAs RDV and NTV. (**a**–**e**) Viral titers in% of DMSO. DMSO (0 µM) was arbitrarily set to 100% (see also Appendix A).

**Figure 3 pharmaceutics-14-01776-f003:**
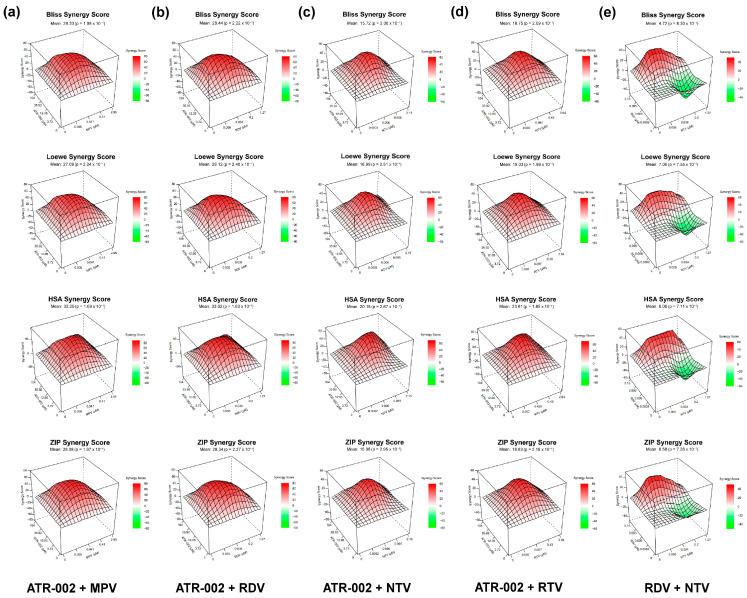
Synergistic evaluation of ATR-002 with DAAs in Calu-3 cells. Landscape visualization of the synergy evaluation shown in Figure 2. ATR-002 was combined with (**a**) MPV, (**b**) RDV, (**c**) NTV or (**d**) RTV. Additionally, the two DAAs, RDV and NTV, were tested in combinational treatment (**e**). Bliss independence (Bliss), Loewe additivity (Loewe), highest single agent (HSA) and Zero Interaction Potency (ZIP) reference models were used to calculate and visualize synergistic areas. Surface is color-coded. Red indicates synergistic interactions and green indicates antagonistic interactions (see also Appendix A).

**Figure 4 pharmaceutics-14-01776-f004:**
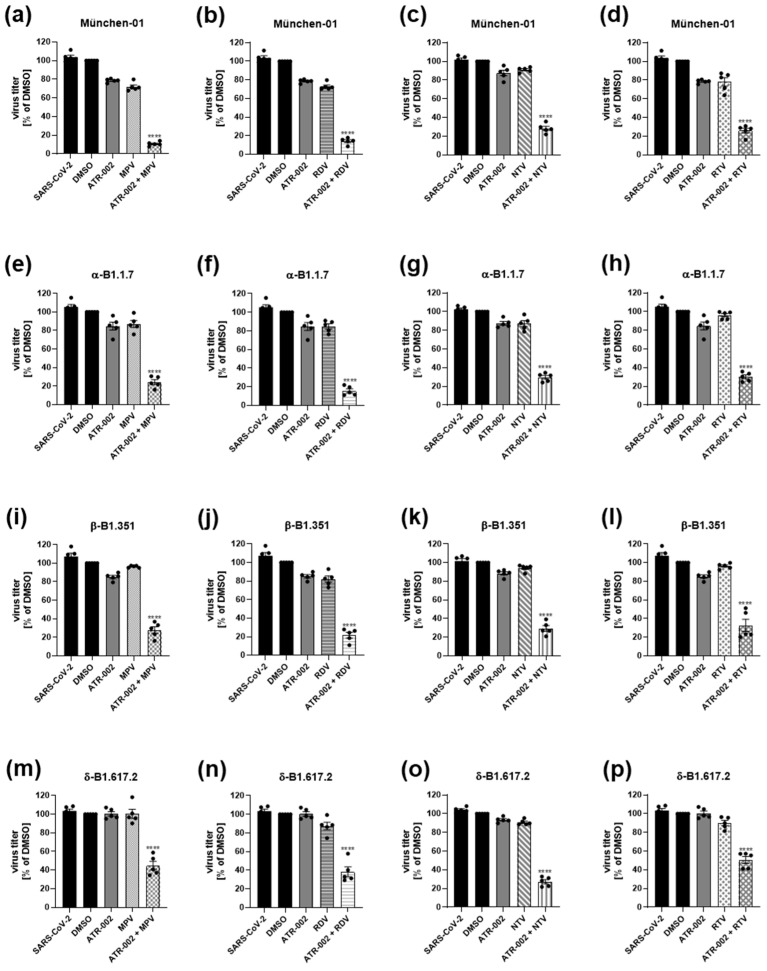
Synergistic effects of MPV, RDV, NTV and RTV in combination with ATR-002 on different SARS-CoV-2 VOCs in Calu-3 cells. Calu-3 cells were infected with SARS-CoV-2 (MOI: 0.01). Drug treatments were initiated 1 h.p.i. MPV (0.047 µM) (**a**,**e**,**i**,**m**), RDV (0.034 µM) (**b**,**f**,**j**,**n**), NTV (0.006 µM) (**c**,**g**,**k**,**o**) or RTV (0.057 µM) (**d**,**h**,**l**,**p**) were combined with ATR-002 (12.05 µM). Untreated (SARS-CoV-2), DMSO-treated and single-treated (drug + DMSO) cells served as controls. Viral titers were analyzed 48 h.p.i. and are depicted in % of DMSO. DMSO was arbitrarily set to 100%. Shown are means ± SEM of five independent experiments, each performed in triplicates. Significance was calculated using an unpaired *t*-test for each inhibitor individually, comparing the single-treated (MPV, RDV, NTV, RTV) vs. the double-treated results (**** *p* ≤ 0.0001) (see also Appendix A).

## Data Availability

Data are contained within the article or Appendix A.

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
