# Peer review of "The MEK1/2 Inhibitor ATR-002 (Zapnometinib) Synergistically Potentiates the Antiviral Effect of Direct-Acting Anti-SARS-CoV-2 Drugs"

_pharmaceutics, 2022, doi:10.3390/pharmaceutics14091776_

Round 1

Reviewer 1 Report

The authors adequately addressed my comments on the previous version. They did several new experiments, which is highly appreciated. The addition of these extra findings significantly increases the relevance of the manuscript.

Therefore, I have no further comments on the revised version. 

Author Response

Reviewer 1: (original reviewers comments in italic)

The authors adequately addressed my comments on the previous version. They did several new experiments, which is highly appreciated. The addition of these extra findings significantly increases the relevance of the manuscript.

Therefore, I have no further comments on the revised version. 

RE:

We thank reviewer 1 for revising our manuscript.

Reviewer 2 Report

The manuscript of Schreber et al. “The MEK1/2 inhibitor ATR-002 (zapnometinib) synergistically potentiates the antiviral effect of direct-acting anti-SARS-CoV-2 drugs” decsribes the synergistic anti-viral effect of direct antivirals against SARS-CoV-2 and host-targeted drug zapnometinib inhibiting Raf/MEK/ERK signaling pathway. The results clearly demonstrate that ATR-002 potentiates the efficacy of both viral polymerase and protease inhibitors due to synergistic effect. The results obtained substantiate combined application of direct antivirals and host-targeted viral inhibitors in the course of coronavirus infection treatment.

The set of methods chosen is adequate, the results are clearly presented. The manuscript can be published in its present form.

Author Response

Reviewer 2: (original reviewers comments in italic)

The manuscript of Schreber et al. “The MEK1/2 inhibitor ATR-002 (zapnometinib) synergistically potentiates the antiviral effect of direct-acting anti-SARS-CoV-2 drugs” decsribes the synergistic anti-viral effect of direct antivirals against SARS-CoV-2 and host-targeted drug zapnometinib inhibiting Raf/MEK/ERK signaling pathway. The results clearly demonstrate that ATR-002 potentiates the efficacy of both viral polymerase and protease inhibitors due to synergistic effect. The results obtained substantiate combined application of direct antivirals and host-targeted viral inhibitors in the course of coronavirus infection treatment.

The set of methods chosen is adequate, the results are clearly presented. The manuscript can be published in its present form.

RE:

We thank reviewer 2 for considering our results clearly presented and our methods adequately chosen.

Reviewer 3 Report

The authors demonstrated that the combination of MEK1/2 inhibitor ATR-002 (zapnometinib) and one of the four antiviral drugs showed synergistic effect against SARS-CoV-2 in vitro. The results are encouraging though more work needs to be done in vivo as the authors pointed out in the discussion section. The manuscript was well written. However, I found the results presented in Figure 4 and Figure S8 don’t seem to agree with each other. Compared to Figure S8, Figure 4 showed higher antiviral effects with higher statistical significance though they were generated from the same set of raw data. I recommend the manuscript be reconsidered for publication after revision.

Author Response

Reviewer 3: (original reviewers comments in italic)

The authors demonstrated that the combination of MEK1/2 inhibitor ATR-002 (zapnometinib) and one of the four antiviral drugs showed synergistic effect against SARS-CoV-2 in vitro. The results are encouraging though more work needs to be done in vivo as the authors pointed out in the discussion section. The manuscript was well written. However, I found the results presented in Figure 4 and Figure S8 don’t seem to agree with each other. Compared to Figure S8, Figure 4 showed higher antiviral effects with higher statistical significance though they were generated from the same set of raw data. I recommend the manuscript be reconsidered for publication after revision.

RE:

We thank reviewer 3 for considering our manuscript well written and to recommend it for publication after the revision.

Reviewer 3 point 1:

I found the results presented in Figure 4 and Figure S8 don’t seem to agree with each other. Compared to Figure S8, Figure 4 showed higher antiviral effects with higher statistical significance though they were generated from the same set of raw data.

RE:

The described putative discrepancy is caused by the different representations of the y-axis. In Figure 4 we do show the relative antiviral effect as % of DMSO (linear scale), whereas in Figure S8, we do show the absolute viral titers as plaque forming units per milliliter (PFU/ml) (logarithmic scale).

Reviewer 4 Report

The authors test a treatment strategy against SARS-CoV-2 infection using the host-targeted agent zapnometinib, or ATR-002, a MEK1/2 inhibitor, in conjunction with the inhibitors of viral RNA-dependent RNA-polymerases Molnupiravir (MPV) and Remdesivir (RDV) or the 3C-like protease inhibitors Nirmatrelvir (NTV) and Ritonavir (RTV). Experiments were performed in vitro in Calu-3 cells using cell viability (as measured by MTT assay) and viral titers (measured by plaque assay) as readouts of drug efficacy. They found that ATR-002 had synergistic effects with the antivirals tested, leading to reduced virus titers, although this effect was somewhat virus strain specific. Overall the manuscript is well written and the conclusions are supported by the data.

There is little introduction as to the rationale of this project. Why is it necessary to test a combination treatment with ATR-002? What advantage does this impart if ATR-002 is an effective therapeutic agent in its own right?

Mechanistic insight into this synergistic effect is lacking. The authors know the mechanism of action of ATR-002. Is this effect altered by the use of the antivirals in conjunction with ATR-002? There is no data or discussion on this, both of which would strengthen the study.

Other comments;

Figures 3 & 4; the legends do not adequately describe what each panel shows, reference should be made to them in the legends.

Figure 4 – suggest including the name of the drug used in the titles of the figures for ease of reference. Further, the dot points cannot be seen in the SARS-CoV-2 and DMSO data due to the black bars. Suggest using box and whisker plots or unfilled bars.

Author Response

Reviewer 4: (original reviewers comments in italic)

The authors test a treatment strategy against SARS-CoV-2 infection using the host-targeted agent zapnometinib, or ATR-002, a MEK1/2 inhibitor, in conjunction with the inhibitors of viral RNA-dependent RNA-polymerases Molnupiravir (MPV) and Remdesivir (RDV) or the 3C-like protease inhibitors Nirmatrelvir (NTV) and Ritonavir (RTV). Experiments were performed in vitro in Calu-3 cells using cell viability (as measured by MTT assay) and viral titers (measured by plaque assay) as readouts of drug efficacy. They found that ATR-002 had synergistic effects with the antivirals tested, leading to reduced virus titers, although this effect was somewhat virus strain specific. Overall the manuscript is well written and the conclusions are supported by the data.

There is little introduction as to the rationale of this project. Why is it necessary to test a combination treatment with ATR-002? What advantage does this impart if ATR-002 is an effective therapeutic agent in its own right?

Mechanistic insight into this synergistic effect is lacking. The authors know the mechanism of action of ATR-002. Is this effect altered by the use of the antivirals in conjunction with ATR-002? There is no data or discussion on this, both of which would strengthen the study.

Other comments;

Figures 3 & 4; the legends do not adequately describe what each panel shows, reference should be made to them in the legends.

Figure 4 – suggest including the name of the drug used in the titles of the figures for ease of reference. Further, the dot points cannot be seen in the SARS-CoV-2 and DMSO data due to the black bars. Suggest using box and whisker plots or unfilled bars.

RE:

We thank Reviewer 4 considering our manuscript well written and our conclusions supported by our data.

Reviewer 4 point 1:

There is little introduction as to the rationale of this project. Why is it necessary to test a combination treatment with ATR-002? What advantage does this impart if ATR-002 is an effective therapeutic agent in its own right?

RE:

We are thankful for this comment. We have added an explanation why it is relevant to test drug synergy between the HTA ATR-002 and different DAAs in the Discussion section (lines 205-210).

Reviewer 4 point 2:

Mechanistic insight into this synergistic effect is lacking. The authors know the mechanism of action of ATR-002. Is this effect altered by the use of the antivirals in conjunction with ATR-002? There is no data or discussion on this, both of which would strengthen the study.

RE:

We thank the reviewer for this comment. We did not extensively discuss this in the previous version of the manuscript because it seemed to us highly speculative. However, we agree with the reviewer that some information on putative mechanisms would be helpful for the reader and now have included a section in the discussion dealing with putative mechanistic explanations of the synergistic effects (lines 221-234).

Reviewer 4 point 3:

Figures 3 & 4; the legends do not adequately describe what each panel shows, reference should be made to them in the legends.

RE:

We thank reviewer 3 for this suggestion. We have included references of the figures panels for Fig. 3 & 4.

Reviewer 4 point 4:

Figure 4 – suggest including the name of the drug used in the titles of the figures for ease of reference. Further, the dot points cannot be seen in the SARS-CoV-2 and DMSO data due to the black bars. Suggest using box and whisker plots or unfilled bars.

RE:

The drugs names were added and the SARS-CoV-2 and DMSO black bars were changed to unfilled bars in figure 4 and supplementary figure 8.

Reviewer 5 Report

Through the submitted manuscript titled “The MEK1/2 inhibitor ATR-002 (zapnometinib) synergistically potentiates the antiviral effect of direct-acting anti-SARS-CoV-2 drugs,” the authors managed to document the importance and applicability of combinatorial therapy over monotherapy when it comes to the management of COVID-19. Experiments were conducted adequately with appropriate replications and controls. Results were properly presented with figures that support the conclusions.  

Authors should consider incorporating CC50 values for the monotherapy and combinatorial therapy experiments in the relevant results sections or figures. 

Author Response

Reviewer 5: (original reviewers comments in italic)

Through the submitted manuscript titled “The MEK1/2 inhibitor ATR-002 (zapnometinib) synergistically potentiates the antiviral effect of direct-acting anti-SARS-CoV-2 drugs,” the authors managed to document the importance and applicability of combinatorial therapy over monotherapy when it comes to the management of COVID-19. Experiments were conducted adequately with appropriate replications and controls. Results were properly presented with figures that support the conclusions.  

Authors should consider incorporating CC50 values for the monotherapy and combinatorial therapy experiments in the relevant results sections or figures. 

RE:

We thank reviewer 5 for evaluating our experiments conducted in an adequate way with appropriate replications and controls, as well as our results properly presented.

Reviewer 5, point 1:

Authors should consider incorporating CC50 values for the monotherapy and combinatorial therapy experiments in the relevant results sections or figures. 

RE:

We thank reviewer 5 for the suggestion. We have carefully considered this option, however, came to the conclusion that the CC50 evaluation is not a main finding of this work. Thus, incorporation of these data to the main body of the manuscript would somehow detract the reader from the important findings. Thus, we decided to leave it in the supplementary data, however, clearly referenced the data in the main text.

This manuscript is a resubmission of an earlier submission. The following is a list of the peer review reports and author responses from that submission.

Round 1

Reviewer 1 Report

pharmaceutics-1684117

The authors report synergistic activity between the MEK1/2 inhibitor ATR-002 and three direct-acting antivirals for SARS-CoV-2 with a more established mechanism of action (i.e. targeting the viral polymerase or 3C protease). The anti-SARS-CoV-2 activity of the MEK1/2 inhibitor by itself was reported by the authors earlier (ref 11). In this follow-up study, the authors collected a broad range of data points, then analyzed these data with synergy analysis software employing four different mathematical models.

Specific comments:

1) Though the synergy score is impressive, it is hard to place this score in a broader context. It would have been very relevant to see how combining a polymerase and protease inhibitor performs in terms of synergy. In other words: is the MEK1/2 inhibitor ‘special’ in this regard?

2) A reduction of 50% in a virus titer of 10exp7 PFU/ml is not meaningful. One wants to see a reduction of 2- to 3-log10 (= 100 to 1000 fold) in virus titer. Since the 50% reduction criterion is used throughout the manuscript, the data are not convincing.

3) The authors should have used nirmatrelvir as a 3CL protease inhibitor. Ritonavir is added in Paxlovid to improve the pharmacokinetics of nirmatrelvir.

4) There is too much redundancy in data presentation. Fig. 1: delete entirely. Fig. 2: for each row (cpd), only two panels need to be kept: the virus titer bar diagram (a, e, …) and dose-response curve (d, f, …). EC1 is a parameter that I have never seen before, and is plausibly very unreliable (only 1% inhibition!?); therefore, delete it entirely. Though the legend of Fig. 2 is very detailed, it does not mention the number of independent experiments.

5) Also Figs 3, 4 and 5 are terribly repetitive, and can easily be combined by removing all the redundant panels. Suggestion: keep only panels a and from e, only the right (3D) panels.

6) I do not see the point of showing the four synergy calculation results if no background is given regarding the pharmacological models behind these Bliss, Loewe, … methods.  

6) Fig. 6: delete the middle panels.

7) The discussion is too long considering that the manuscript contains only one message. Instead, explain WHY the MEK1/2 inhibitor potentiates the activity of direct antivirals.

Reviewer 2 Report

The manuscript entitled “The MEK1/2 inhibitor ATR-002 (zapnometinib) synergistically potentiates the antiviral effect of Molnupiravir, Remdesivir and Ritonavir against SARS-CoV-2” by Schreiber et. al. shows that the MEK1/2 inhibitor ATR-002 exerts synergistic antiviral effects against SARS-CoV-2 when combined with these three drugs.

The paper is well written and the data is clearly presented.

There are only a few remarks:

  • in Figure 2, the number of replicates of the virus titration is missing and should be mentioned
  • Protocol for virus infection in the presence of inhibitors should be described in more detail in the M&M section, e.g. Calu-3 cell seeding (number of cells, well size), was there a wash of cells after virus infection and before compound treatment?
  • It would be of interest to know what the assumptions are for the different synergy models used in this study
  • I would call Figure 1 rather a Table than a Figure. It would also fit better to the end of the manuscript as it summarizes the findings shown in the figures and should support the conclusions.